META-RESEARCH ARTICLE

# Tracking changes between preprint posting and journal publication during a pandemic

**Liam Brierley**[1], **Federico Nanni**[2☯], **Jessica K. Polka**[3☯], **Gautam Dey**[4], **Máté Pálfy**[5], **Nicholas Fraser**[6], **Jonathon Alexis Coates**[7]*

**1** Department of Health Data Science, University of Liverpool, Liverpool, United Kingdom, **2** The Alan Turing Institute, London, United Kingdom, **3** ASAPbio, San Francisco, California, United States of America, **4** Cell Biology and Biophysics Unit, European Molecular Biology Laboratory, Heidelberg, Germany, **5** The Company of Biologists, Histon, Cambridge, United Kingdom, **6** Leibniz Information Centre for Economics, Leibniz, Germany, **7** William Harvey Research Institute, Barts and the London School of Medicine and Dentistry Queen Mary University of London, London, United Kingdom

☯ These authors contributed equally to this work.

\* jonathon.coates@qmul.ac.uk

**Data Availability Statement:** All data and code is available from: https://github.com/preprinting-a-pandemic/preprint_changes and https://zenodo.org/record/5594903#.YXUv9_nTUuU.

## Abstract

Amid the Coronavirus Disease 2019 (COVID-19) pandemic, preprints in the biomedical sciences are being posted and accessed at unprecedented rates, drawing widespread attention from the general public, press, and policymakers for the first time. This phenomenon has sharpened long-standing questions about the reliability of information shared prior to journal peer review. Does the information shared in preprints typically withstand the scrutiny of peer review, or are conclusions likely to change in the version of record? We assessed preprints from bioRxiv and medRxiv that had been posted and subsequently published in a journal through April 30, 2020, representing the initial phase of the pandemic response. We utilised a combination of automatic and manual annotations to quantify how an article changed between the preprinted and published version. We found that the total number of figure panels and tables changed little between preprint and published articles. Moreover, the conclusions of 7.2% of non-COVID-19–related and 17.2% of COVID-19–related abstracts undergo a discrete change by the time of publication, but the majority of these changes do not qualitatively change the conclusions of the paper.

## Introduction

Global health and economic development in 2020 were overshadowed by the Coronavirus Disease 2019 (COVID-19) pandemic, which grew to over 3.2 million cases and 220,000 deaths within the first 4 months of the year [1–3]. The global health emergency created by the pandemic has demanded the production and dissemination of scientific findings at an unprecedented speed via mechanisms such as preprints, which are scientific manuscripts posted by their authors to a public server prior to the completion of journal-organised peer review [4–6]. Despite a healthy uptake of preprints by the bioscience communities in recent years [7], some concerns persist [8–10]. In particular, one such argument suggests that preprints are less

**Funding:** NF acknowledges funding from the German Federal Ministry for Education and Research, grant numbers 01PU17005B (OASE) and 01PU17011D (QuaMedFo). LB acknowledges funding from a Medical Research Council Skills Development Fellowship award, grant number MR/T027355/1. GD thanks the European Molecular Biology Laboratory for support. The funders had no role in study design, data collection and analysis, decision to publish, or preparation of the manuscript.

**Competing interests:** I have read the journal's policy and the authors of this manuscript have the following competing interests: JP is the executive director of ASAPbio, a non-profit organization promoting the productive use of preprints in the life sciences. GD is a bioRxiv Affiliate, part of a volunteer group of scientists that screen preprints deposited on the bioRxiv server. GD and JAC are contributors to preLights and ASAPbio Fellows.

**Abbreviations:** API, Application Programming Interface; CI, confidence interval; COVID-19, Coronavirus Disease 2019; LRT, likelihood ratio test; NLP, natural language processing; RCT, randomised controlled trial; VIF, variance inflation factor.

reliable than peer-reviewed papers, since their conclusions may change in a subsequent version. Such concerns have been amplified during the COVID-19 pandemic, since preprints are being increasingly used to shape policy and influence public opinion via coverage in social and traditional media [11,12]. One implication of this hypothesis is that the peer review process will correct many errors and improve reproducibility, leading to significant differences between preprints and published versions.

Several studies have assessed such differences. For example, Klein and colleagues used quantitative measures of textual similarity to compare preprints from arXiv and bioRxiv with their published versions [13], concluding that papers change "very little." Recently, Nicholson and colleagues employed document embeddings to show that preprints with greater textual changes compared with the journal versions took longer to be published and were updated more frequently [14]. However, changes in the meaning of the content may not be directly related to changes in textual characters, and vice versa (e.g., a major rearrangement of text or figures might simply represent formatting changes, while the position of a single decimal point could significantly alter conclusions). Therefore, sophisticated approaches aided or validated by manual curation are required, as employed by 2 recent studies. Using preprints and published articles, both paired and randomised, Carneiro and colleagues employed manual scoring of methods sections to find modest, but significant improvements in the quality of reporting among published journal articles [15]. Pagliaro manually examined the full text of 10 preprints in chemistry, finding only small changes in this sample [16], and Kataoka compared the full text of medRxiv randomised controlled trials (RCTs) related to COVID-19, finding in preprint versions an increased rate of spin (positive terms in the title or abstract conclusion section used to describe nonsignificant results [17]. Bero and colleagues [18] and Oikonomidi and colleagues [19] investigated changes in conclusions reported in COVID-19–related clinical studies, finding that some preprints and journal articles differed in the outcomes reported. However, the frequency of changes in the conclusions of a more general sample of preprints remained an open question. We sought to identify an approach that would detect such changes effectively and without compromising on sample size. We divided our analysis between COVID-19 and non-COVID-19 preprints, as extenuating circumstances such as expedited peer review and increased attention [11] may impact research related to the pandemic.

To investigate how preprints have changed upon publication, we compared abstracts, figures, and tables of bioRxiv and medRxiv preprints with their published counterparts to determine the degree to which the top-line results and conclusions differed between versions. In a detailed analysis of abstracts, we found that most scientific articles undergo minor changes without altering the main conclusions. While this finding should provide confidence in the utility of preprints as a way of rapidly communicating scientific findings that will largely stand the test of time, the value of subsequent manuscript development, including peer review, is underscored by the 7.2% of non-COVID-19–related and 17.2% of COVID-19–related preprints with major changes to their conclusions upon publication.

## Results

### COVID-19 preprints were rapidly published during the early sphase of the pandemic

The COVID-19 pandemic has spread quickly across the globe, reaching over 3.2 million cases worldwide within 4 months of the first reported case [1]. The scientific community responded concomitantly, publishing over 16,000 articles relating to COVID-19 within 4 months [11]. A large proportion of these articles (>6,000) were manuscripts hosted on preprint servers. Following this steep increase in the posting of COVID-19 research, traditional publishers adapted

new policies to support the ongoing public health emergency response efforts, including efforts to fast-track peer review of COVID-19 manuscripts (e.g., *eLife* [20]). At the time of our data collection in May 2020, 4.0% of COVID-19 preprints were published by the end of April, compared to the 3.0% of non-COVID-19 preprints that were published such that we observed a significant association between COVID-19 status (COVID-19 or non-COVID-19 preprint) and published status (chi-squared test; $\chi^2$ = 6.78, df = 1, $p$ = 0.009, $n$ = 14,812) (Fig 1A). When broken down by server, 5.3% of COVID-19 preprints hosted on bioRxiv were published compared to 3.6% of those hosted on medRxiv (S1A Fig). However, a greater absolute number of medRxiv versus bioRxiv COVID-19 preprints (71 versus 30) were included in our sample of detailed analysis of text changes (see Methods), most likely a reflection of the different focal topics between the 2 servers (medRxiv has a greater emphasis on medical and epidemiological preprints).

A major concern with expedited publishing is that it may impede the rigour of the peer review process [21]. Assuming that the version of the manuscript originally posted to the preprint server is likely to be similar to that subjected to peer review, we looked to journal peer review reports to reveal reviewer perceptions of submitted manuscripts. For our detailed sample of $n$ = 184 preprint–published article pairs, we assessed the presence of transparent peer review (defined as openly available peer review reports published by the journal alongside the article; we did not investigate the availability of non-journal peer reviews of preprints) and found that only a minority of preprints that were subsequently published were associated with transparent journal reviews, representing 3.4% of COVID-19 preprints and 12.4% of non-COVID-19 preprints examined, although we did not observe strong evidence of an association between COVID-19 status and transparent peer review ($\chi^2$ = 3.76, df = 1, $p$ = 0.053)) (Fig 1B). The lack of transparent peer reviews was particularly apparent for research published from medRxiv (S1B Fig). Data availability is a key component of the open science initiative, but journal policies differ in the requirement for open data. Moreover, evidence suggests that non-scientists are utilising underlying raw data to promote misinformation [22]; we therefore investigated the availability of underlying data associated with preprint–published article pairs. There was little difference in data availability between the preprint and published version of an article. Additionally, we found no evidence of association between overall data availability and COVID-19 status (Fisher exact, 1,000 simulations; $p$ = 0.583). However, we note that a greater proportion of COVID-19 articles had a reduction in data availability when published (4.6% versus 2.1%), and vice versa, a greater proportion of non-COVID-19 articles were more likely to have additional data available upon publishing (20.6% versus 12.6%) (Fig 1C). This trend was reflected when broken down by preprint server (S1C Fig).

The number of authors may give an indication of the amount of work involved; we therefore assessed authorship changes between the preprint and published articles. Although the vast majority (>85%) of preprints did not have any changes in authorship when published (Fig 1D), we found weak evidence of association between authorship change (categorised as any versus none) and COVID-19 status ($\chi^2$ = 3.90, df = 1, $p$ = 0.048). Specifically, COVID-19 preprints were almost 3 times as likely to have additional authors (categorised as any addition versus no additions) when published compared to non-COVID-19 preprints (17.2% versus 6.2%) ($\chi^2$ = 4.51, df = 1, $p$ = 0.034). When these data were broken down by server, we found that none of the published bioRxiv preprints had any author removals or alterations in the corresponding author (S1D Fig).

Having examined the properties of preprints that were being published within our time frame, we next investigated which journals were publishing these preprints. Among our sample of published preprints, those describing COVID-19 research were split across many journals, with clinical or multidisciplinary journals tending to publish the most papers that were

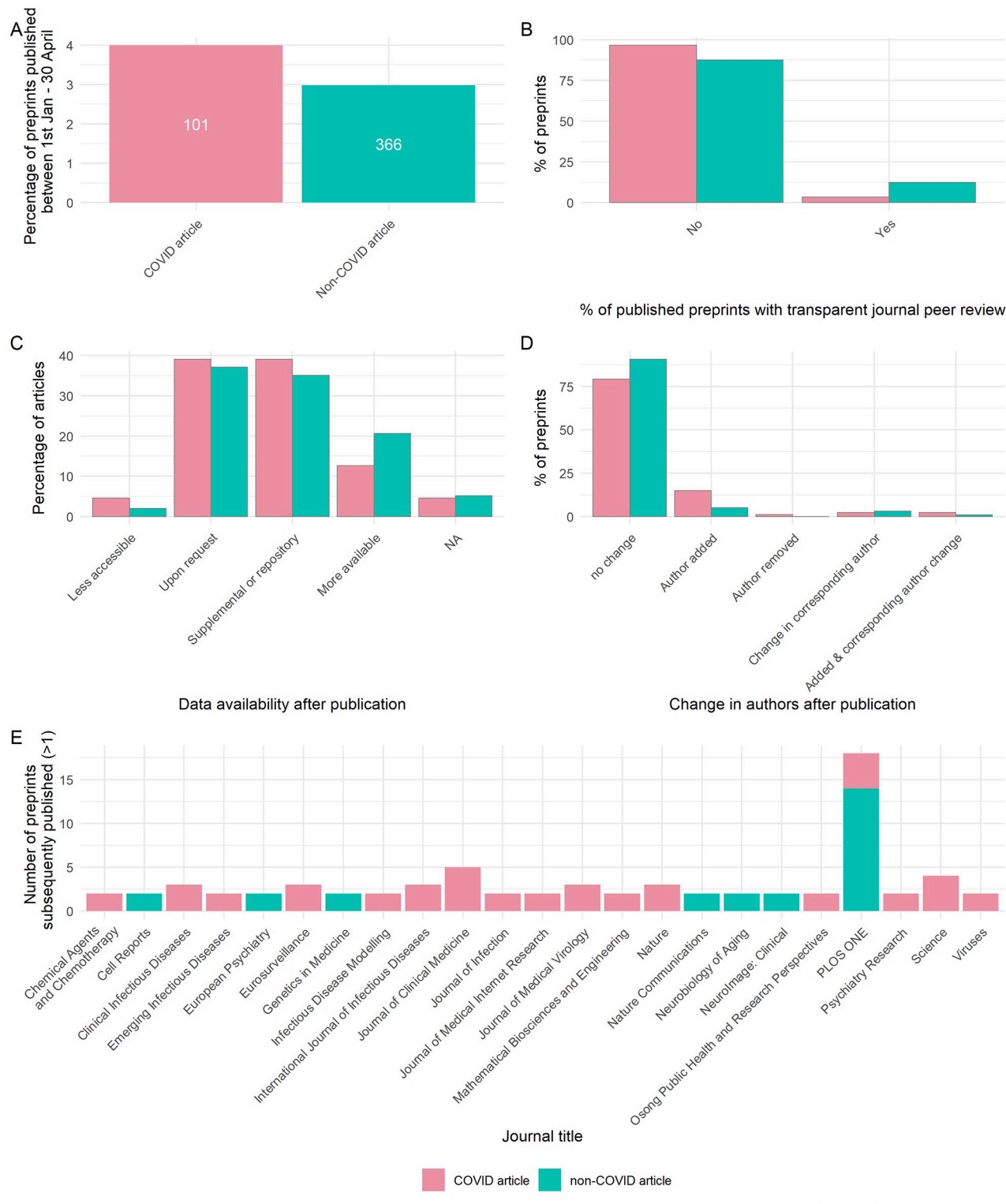

**Fig 1. Publishing and peer review of preprints during the COVID-19 pandemic.** (**A**) Percentage of COVID-19 and non-COVID-19 preprints published by April 30, 2020. Labels denote absolute number. (**B**) Percentage of published preprints associated with transparent peer review (the publication of review reports with the journal version of the article). (**C**) Data availability after publication. (**D**) Change in authorship after publication. (**E**) Journals that are publishing preprints. Panel A describes all available data (*n* = 14,812 preprints), while panels B–E describe sample of preprints analysed in detail (*n* = 184). The data underlying this figure may be found at https://github.com/preprinting-a-pandemic/preprint_changes and https://zenodo.org/record/5594903#.YXUv9_nTUuU. COVID-19, Coronavirus Disease 2019.

previously preprints (Fig 1E). Non-COVID-19 preprints were mostly published in *PLOS ONE*, although they were also found in more selective journals, such as *Cell Reports*. When broken down by server, preprints from bioRxiv were published in a range of journals, including the highly selective *Nature* and *Science* (S1E and S1F Fig); interestingly, these were all COVID-19 articles. Together, these data reveal that preprints are published in diverse venues and suggest that during the early phase of the pandemic, COVID-19 preprints were being expedited through peer review compared to non-COVID-19 preprints. However, published articles were rarely associated with transparent peer review, and 38% of the literature sampled had limited data availability, with COVID-19 status having little impact on these statistics.

## Figures do not majorly differ between the preprint and published version of an article

One proxy for the total amount of work, or number of experiments, within an article is to quantify the number of panels in each figure [23]. We therefore quantified the number of panels and tables in each article in our dataset.

We found that, on average, there was no difference in the total number of panels and tables between the preprint and published version of an article. However, COVID-19 articles had fewer total panels and tables compared to non-COVID-19 articles (Mann–Whitney; median (IQR) = 7 (6.25) versus 9 (10) and $p$ = 0.001 for preprints, median (IQR) = 6 (7) versus 9 (10) and $p$ = 0.002 for published versions) (Fig 2A). For individual preprint–published pairs, we found comparable differences in numbers of panels and tables for COVID-19 and non-COVID-19 articles (Fig 2B). Preprints posted to bioRxiv contained a higher number of total panels and tables (Mann–Whitney; $p < 0.001$ for both preprints and their published versions) and greater variation in the difference between the preprint and published articles than preprints posted to medRxiv (Fligner–Killeen; $\chi^2$ = 9.41, df = 1, $p$ = 0.002) (S2A and S2B Fig).

To further understand the types of panel changes, we classified the changes in panels and tables as panels being added, removed, or rearranged. Independent of COVID-19-status, over 75% of published preprints were classified with "no change" or superficial rearrangements to panels and tables, confirming the previous conclusion. Despite this, approximately 23% of articles had "significant content" added or removed from the figures between preprint and final versions (Fig 2C). None of the preprints posted to bioRxiv experienced removal of content upon publishing (S2C Fig).

These data suggest that, for most papers in our sample, the individual panels and tables do not majorly change upon journal publication, suggesting that there are limited new experiments or analyses when publishing previously posted preprints.

We found no discernible pattern in the degree to which figures changed based on the destination journal of either COVID-19 (Fig 2D) or non-COVID-19 papers (Fig 2E), although the latter were distributed among a larger range of journals.

## The majority of abstracts do not discretely change their main conclusions between the preprint and published article

We compared abstracts between preprints and their published counterparts that had been published in the first 4 months of the COVID-19 pandemic (January to April 2020 with an extended window for non-COVID-19 articles of September 2019 to April 2020). Abstracts contain a summary of the key results and conclusions of the work and are freely accessible; they are the most read section. To computationally identify all individual changes between the preprint and published versions of the abstract and derive a quantitative measure of similarity between the 2, we applied a series of well-established string-based similarity scores, already

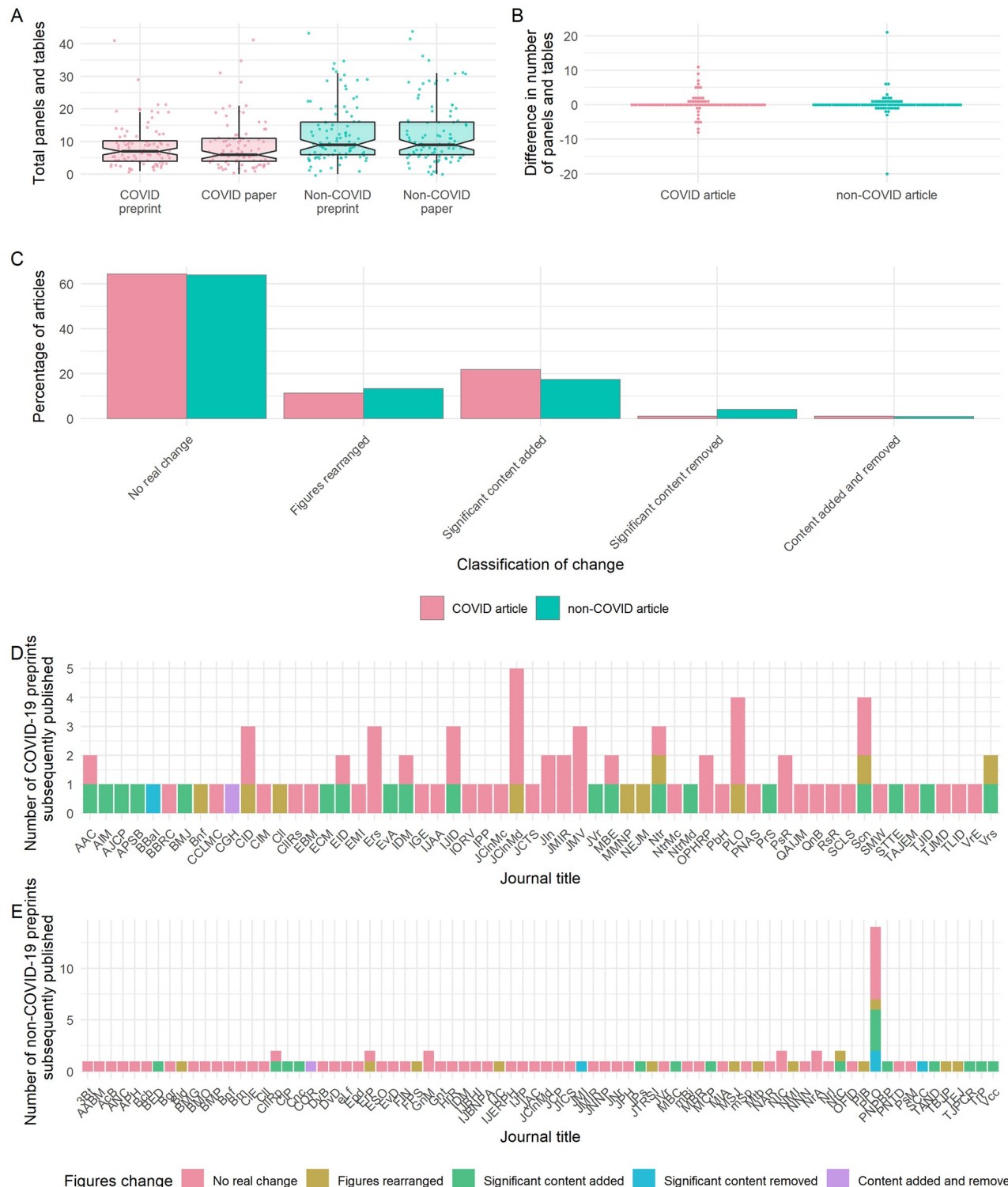

**Fig 2. Preprint–publication pairs do not significantly differ in the total numbers of panels and tables. (A)** Total numbers of panels and tables in preprints and published articles. Boxplot notches denote approximated 95% CI around medians. **(B)** Difference in the total number of panels and tables between the preprint and published versions of articles. **(C)** Classification of figure changes between preprint and published articles. **(D)** Journals publishing COVID-19 preprints, based on annotated changes in panels. **(E)** Journals publishing non-COVID-19 preprints, based on annotated changes in panels. All panels describe sample of preprints analysed in detail ($n$ = 184). See S1 Text for key to abbreviated journal labels. The data underlying this figure may be found at https://github.com/preprinting-a-pandemic/preprint_changes and https://zenodo.org/record/5594903#.YXUv9_nTUuU. CI, confidence interval; COVID-19, Coronavirus Disease 2019.

validated to work for such analyses. We initially employed the python SequenceMatcher (dif-flib module), based on the "Gestalt Pattern Matching" algorithm [24], which determines a change ratio by iteratively aiming to find the longest contiguous matching subsequence given 2 pieces of text. We found that COVID-19 abstracts had a significantly greater change ratio than non-COVID-19 abstracts (Mann–Whitney; median (IQR) = 0.338 (0.611) versus 0.197 (0.490) and $p$ = 0.010), with a sizeable number ($n$ = 20) appearing to have been substantially rewritten such that their change ratio was ≥0.75 (Fig 3A). However, one limitation of this method is that it cannot always handle rearrangements properly (e.g., a sentence moved from the beginning of the abstract to the end), and these are often counted as changes between the 2 texts. As a comparison to this open-source implementation, we employed the output of the Microsoft Word track changes algorithm and used this as a different type of input for deter-mining the change ratio of 2 abstracts.

Using this method, we confirmed that abstracts for COVID-19 articles changed more than for non-COVID-19 articles (Mann–Whitney; median (IQR) = 0.203 (0.287) versus 0.094 (0.270) and $p$ = 0.007), although the overall degree of changes observed were reduced (Fig 3B); this suggests that while at first look a pair of COVID-19 abstracts may seem very different between their preprint and published version, most of these changes are due to reorganisation of the content. Nonetheless, the output obtained by the Microsoft Word track changes algo-rithm highlights that it is more likely that COVID-19 abstracts undergo larger rewrites (i.e., their score is closer to 1.0).

Since text rearrangements may not result in changes in meaning, 4 annotators indepen-dently annotated the compared abstracts according to a rubric we developed for this purpose (Table 1, S2 Method). We found that independent of COVID-19-status, a sizeable number of abstracts did not undergo any meaningful changes (24.1% of COVID-19 and 36.1% of non-COVID-19 abstracts). Over 50% of abstracts had changes that minorly altered, strengthened, or softened the main conclusions (Fig 3C; see representative examples in S2 Table). A total of 17.2% of COVID-19 abstracts and 7.2% of non-COVID-19 abstracts had major changes in their conclusions. A main conclusion of one of these abstracts (representing 0.5% of all abstracts scored) contradicted its previous version. Excerpts including each of these major changes are listed in S3 Table. Using the degree of change, we evaluated how the manual scor-ing of abstract changes compared with our automated methods. We found that difflib change ratios and Microsoft Word change ratios significantly differed between our manual rating of abstracts based on highest change (Kruskal–Wallis; $p$ < 0.001 in both cases) (S3A and S3B Fig). Specifically, change ratios were significantly greater in abstracts having "minor change" than "no change" (post hoc Dunn test; Bonferroni-adjusted $p$ < 0.001 in both cases), but abstracts having "major change" were only greater than "minor change" for Microsoft Word and not difflib change ratio (Bonferroni-adjusted $p$ = 0.01, 0.06, respectively).

Among annotations that contributed minorly to the overall change of the abstract, we also annotated a neutral, positive, or negative direction of change (Table 1, S2 Method). Most of these changes were neutral, modifying the overall conclusions somewhat without directly strengthening or softening them (see examples in S2 Table). Among changes that strengthened or softened conclusions, we found abstracts that contained only positive changes or only nega-tive changes, and many abstracts displayed both positive and negative changes (Fig 3D), in both COVID-19 and non-COVID-19 articles. When we assessed the sum of positive or nega-tive scores based on the manually rated abstract change degree, we found each score sum (i.e., number of positive or negative scores) significantly differed between ratings (Kruskal–Wallis; $p$ < 0.001 in both cases). Abstracts having "minor change" had greater sum scores than those with "no change" (post hoc Dunn test; Bonferroni-adjusted $p$ < 0.001 in both cases), while abstracts having "major change" had greater sum positive scores than those with "minor

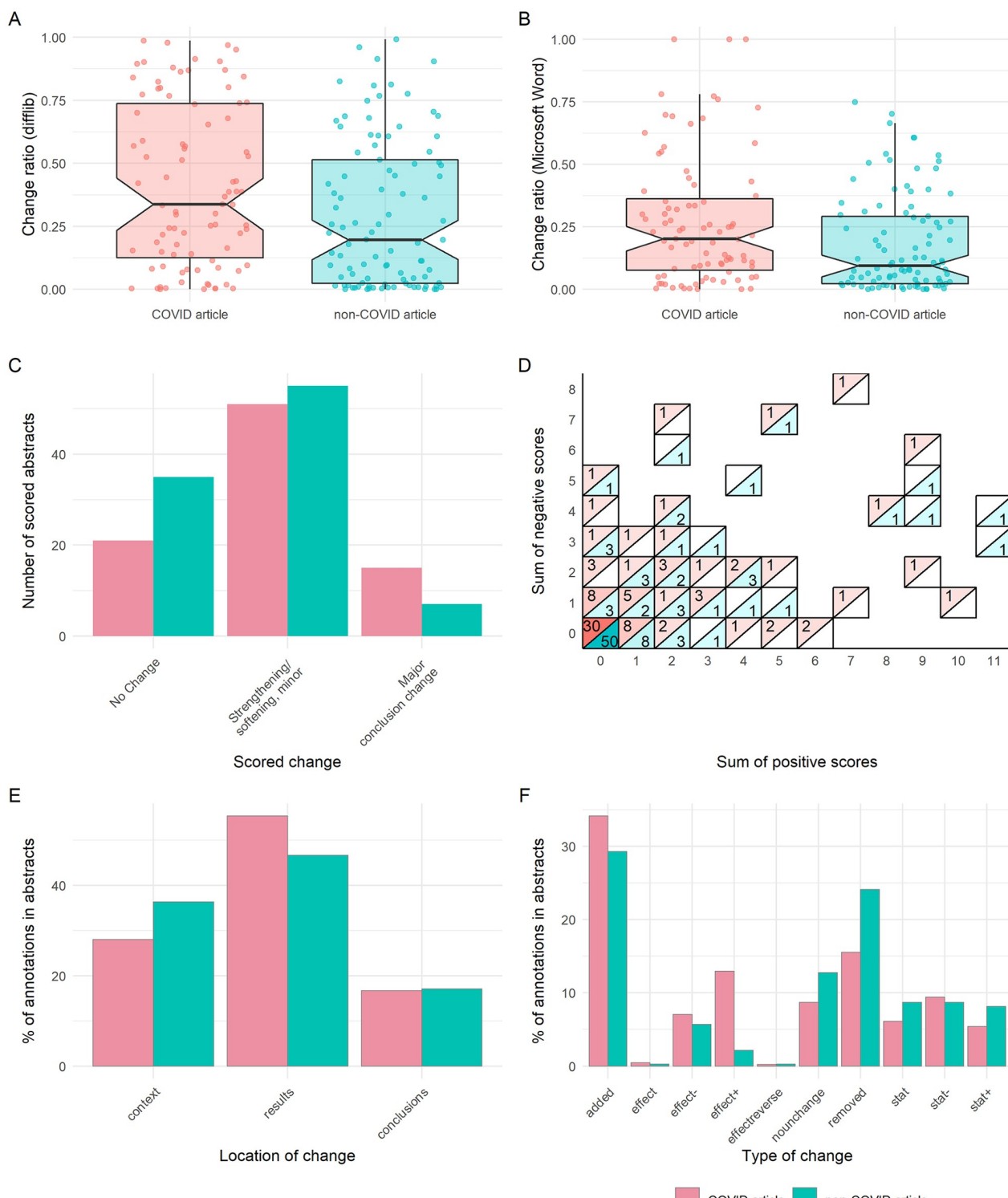

**Fig 3. Preprint–publication abstract pairs have substantial differences in text, but not interpretation. (A)** Difflib calculated change ratio for COVID-19 or non-COVID-19 abstracts. **(B)** Change ratio calculated from Microsoft Word for COVID-19 or non-COVID-19 abstracts. **(C)** Overall changes in abstracts for COVID-19 or non-COVID-19 abstracts. **(D)** Sum of positive and negative annotations for COVID-19 or non-COVID-19 abstracts, with colour and label denoting number of abstracts with each particular sum combination. **(E)** Location of annotations within COVID-19 or non-COVID-19 abstracts. **(F)** Type of annotated change within COVID-19 or non-COVID-19 abstracts. All panels describe sample of abstracts analysed in detail (*n* = 184). Boxplot notches denote approximated 95% CI around medians. The data underlying this figure may be found at https://github.com/preprinting-a-pandemic/preprint_changes and https://zenodo.org/record/5594903#.YXUv9_nTUuU. CI, confidence interval; COVID-19, Coronavirus Disease 2019.

**Table 1. Tags (1 each of section, type, and degree) applied to each annotation of text meaningfully changed in abstracts.**

| Section | Description |
|---|---|
| context | Background or methods |
| results | A statement linked directly to data |
| conclusion | Interpretations and/or implications |
| **Type** | **Description** |
| added | New assertion |
| removed | Assertion removed |
| nounchange | One noun is substituted for another ("fever" becomes "high fever") |
| effectreverse | The opposite assertion is now being made (word "negatively" added) |
| effect+ | The effect is now stronger (changes in verbs/adjectives/adverbs) |
| effect− | The effect is now weaker (changes in verbs/adjectives/adverbs) |
| stat+ | Statistical significance increased (expressed as number or in words) |
| stat− | Statistical significance decreased (expressed as number or in words) |
| statinfo | Addition/removal of statistical information (like a new test or CIs) |
| **Degree** | **Description** |
| 1 | Significant: minorly alters a main conclusion of the paper |
| 1− | Significant: softens a main conclusion of the paper |
| 1+ | Significant: strengthens a main conclusion of the paper |
| 2 | Major: a discrete change in a main conclusion of the paper |
| 3 | Massive: a main conclusion of the paper contradicts its earlier version |

CI, confidence interval.

change," but not greater sum negative scores (Bonferroni-adjusted $p$ = 0.019, 0.329, respectively) (S3C Fig).

We next assessed whether certain subsections of the abstract were more likely to be associated with changes. The majority of changes within abstracts were associated with results, with a greater observed proportion of such annotations for COVID-19 abstracts than non-COVID-19 abstracts (55.3% and 46.6%, respectively) (Fig 3E). We then evaluated the type of change in our annotations, for example, changes to statistical parameters/estimates or addition or removal of information. This demonstrated that the most frequent changes were additions of new findings to the abstracts following peer review, followed by removals, which were more common among non-COVID-19 manuscripts (Fig 3F). We also frequently found an increase in sample sizes or the use/reporting of statistical tests (type "stat+") in the published version of COVID-19 articles compared to their preprints (S2 Table).

We then investigated whether abstracts with minor or major overall changes more frequently contained certain types or locations of changes. We found that abstracts with both major and minor conclusion changes had annotations in all sections, and both degrees of change were also associated with most types of individual changes. For non-COVID-19 abstracts, 80.7% of our annotated changes within conclusion sections and 92.2% of our annotated changes within contexts ($n$ = 46 and 118 annotations, respectively) belonged to abstracts categorised as having only minor changes (S3D Fig). Moreover, the majority of annotated changes in statistics (between 73.9% and 96.7% depending on COVID-19 status and type of change) were within abstracts with minor changes (S3E Fig).

We next examined whether the manually rated degree of abstract change was associated with the delay between preprint posting and journal publication. COVID-19 articles in our annotated sample were published more rapidly (Mann–Whitney; $p < 0.001$), with a median

delay of 19 days (IQR = 15.5), compared to 101 days (IQR = 79) for non-COVID-19 articles (S3F Fig). Although degree of change were not associated with publishing delay for COVID-19 articles (Kruskal–Wallis; $p$ = 0.397), an association was detected for non-COVID-19 articles ($p$ = 0.002). Specifically, non-COVID-19 articles with no change were published faster than those with minor changes (post hoc Dunn test; median (IQR) = 78 days (58) versus 113 days (73), and Bonferroni-adjusted $p < 0.001$) but not faster than those with major changes (median (IQR) = 78 days (58) versus 111 days (42.5) and $p$ = 0.068) (S3F Fig), although we only observed 7 such articles, limiting the interpretation of this finding.

We then investigated which journals were publishing preprints from those with each scored degree of change within our sample (S3G Fig, S1 Table). We found that *PLOS ONE* was the only journal to publish more than 1 preprint that we determined to have major changes in the conclusions of the abstract, although this journal published the most observed non-COVID-19 preprints. Similarly, *PLOS ONE*, *Eurosurveillance*, *Science*, and *Nature* were the only journals observed to published more than 2 preprints that we deemed as having any detectable conclusion changes (major or minor).

Finally, to confirm whether our observed patterns may differ for particular research fields, we examined degree and type of changes for a subgroup of medRxiv preprints. We selected the combined categories of "infectious diseases" ($n$ = 29) and "epidemiology" ($n$ = 28) as the most frequent of the 48 bioRxiv and medRxiv categorisations in our sample, and the categories arguably most generally reflective of COVID-19 research (although 10 of these preprints were non-COVID-19 related). For this subgroup, we confirmed that COVID-19 abstracts had significantly greater difflib and Microsoft Word change ratios than non-COVID-19 abstracts (Mann–Whitney; $p$ = 0.010, 0.007) (S4A and S4B Fig). Again, over 50% of these abstracts were rated as having minor changes and 17.5% rated as having major changes, although these mostly occurred within COVID-19 preprints (S4C Fig). Similar proportions of figure change ratings were also observed (S4D Fig), with a slightly greater proportion of non-COVID-19 preprints having figures rearranged. Locations and types of individual changes also appeared consistent between infectious disease/epidemiology preprints and our full sample, with slightly lower proportions of changes to results and changes involving removed assertions and increased statistical significant for non-COVID-19 preprints (S4E and S4F Fig).

These data reveal that abstracts of preprints mostly experience minor changes prior to publication. COVID-19 articles experienced greater alterations than non-COVID-19 preprints and were slightly more likely to have major alterations to the conclusions. Overall, most abstracts are comparable between the preprinted and published article.

## Changes in abstracts and figures are weakly associated with Twitter attention, comments, and citations

During the COVID-19 pandemic, preprints have received unprecedented attention across social media and in the use of commenting systems on preprint servers [11]. A small proportion of these comments and tweets can be considered as an accessory form of peer review [25]. We therefore next investigated if community commentary was associated with degree of changes to abstracts or figures. Additionally, to determine if the scientific community was detecting any difference in the reliability of the preprints that change upon publication, we also investigated associations between degree of changes and preprint citations.

Initially, we found significant associations between manually categorised degree of change to preprint abstracts and the numbers of tweets, preprint repository comments, and citations (Kruskal–Wallis; $p$ = 0.038, 0.031, 0.008, respectively; Fig 4). However, no associations were detected with degree of changes to figures ($p$ = 0.301, 0.428, 0.421, respectively; Fig 4). We also

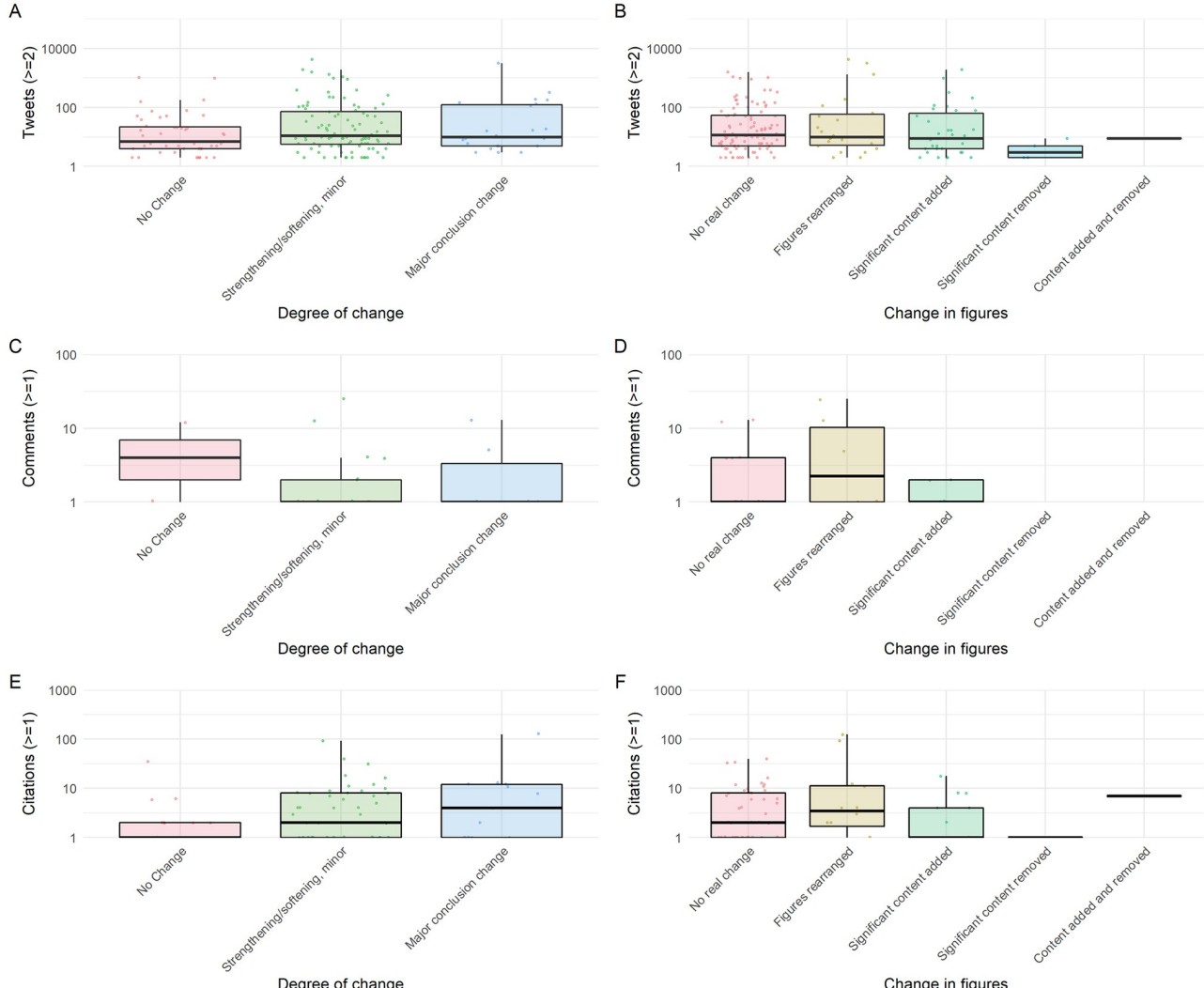

**Fig 4. Altmetric data for overall degree of change in abstracts and figures. (A)** Number of tweets (at least 2) and overall abstract change. **(B)** Number of tweets (at least 2) and overall change in figures. **(C)** Number of comments (at least 1) and overall abstract change. **(D)** Number of comments (at least 1) and overall change in figures. **(E)** Number of preprint citations (at least 1) based on overall abstract change. **(F)** Number of preprint citations (at least 1) based on overall change in figures. The data underlying this figure may be found at https://github.com/preprinting-a-pandemic/preprint_changes and https://zenodo.org/record/5594903#.YXUv9_nTUuU.

found significant weak positive correlations (Spearman rank; $0.133 \leq \rho \leq 0.205$) between each usage metric and automated difflib change ratios ($p = 0.030, 0.009, 0.005$, respectively) and Microsoft Word change ratios, except for number of tweets ($p = 0.071, 0.020, 0.013$, respectively).

When adjusted for COVID-19 status, delay between posting and publication, and total time online in a multivariate regression, several of these associations persisted (Table 2). Compared to preprints with no figure changes, those with rearranged figures were tweeted at almost 3 times the rate (rate ratio = 2.89, 95% CI = [1.54, 5.79]) while those with content added and removed were tweeted much lower rates (rate ratio = 0.11, 95% CI = [0.01, 1.74]). Additionally, preprint abstracts with text changes in published versions substantial enough to reach the maximum difflib change ratio (i.e., 1) had received comments at an estimated 10 times the rate (rate ratio = 9.81, 95% CI = [1.16, 98.41]) and received citations at 4 times the rate (rate

**Table 2. Outputs from multivariate negative binomial regressions predicting counts of usage metrics for 184 preprint–paper pairs.**

| Covariate term | Tweets | | Comments | | Citations | |
|---|---|---|---|---|---|---|
| | LRT | p(LRT) | LRT | p(LRT) | LRT | p(LRT) |
| Degree of abstract change (no change/minor/major) | 3.294 | 0.193 | 0.229 | 0.892 | 3.563 | 0.168 |
| Degree of figure change (no change/rearranged/content added/content removed) | 17.443 | 0.002 | 5.974 | 0.201 | 5.116 | 0.276 |
| Difflib change ratio | 1.272 | 0.259 | 4.392 | 0.036 | 5.564 | 0.018 |
| Microsoft Word change ratio | 1.453 | 0.228 | 1.358 | 0.244 | 3.328 | 0.068 |
| COVID-19 status (COVID-19 or non-COVID-19) | 90.79 | <0.001 | 10.627 | 0.001 | 86.207 | **<0.001** |
| Delay between preprint posting and journal publication (days) | 1.661 | 0.197 | 8.16 | 0.004 | 0.676 | 0.411 |
| Time since posted by end of sampling (days) | 13.264 | <0.001 | 5.596 | 0.018 | 34.675 | **<0.001** |

LRT denotes likelihood ratio test statistic. Bold denotes covariates with $p < 0.05$.

COVID-19, Coronavirus Disease 2019.

ratio = 4.26, 95% CI = [1.27, 14.90]) of preprints with no change (i.e., difflib change ratio = 0). However, among our detailed sample of 184 preprint–paper pairs, only a minority were observed to receive any comments ($n = 28$) or citations at all ($n = 81$), and usage was explained much more strongly by COVID-19 status and time since posted than any measure of change among our sampled pairs (Table 2).

Together, our sampled data suggest that the amount of attention given to a preprint does not reflect or impact how much it will change upon publication, although preprints undergoing discrete textual changes are commented upon and cited more often, perhaps reflecting additional value added by peer review.

## Discussion

With a third of the early COVID-19 literature being shared as preprints [11], we assessed the differences between these preprints and their subsequently published versions and compared these results to a similar sample of non-COVID-19 preprints and their published articles. This enabled us to provide quantitative evidence regarding the degree of change between preprints and published articles in the context of the COVID-19 pandemic. We found that preprints were most often passing into the "permanent" literature with only minor changes to their conclusions, suggesting that the entire publication pipeline is having a minimal but beneficial effect upon preprints (e.g., by increasing sample sizes or statistics or by making author language more conservative) [13,15].

The duration of peer review has drastically shortened for COVID-19 manuscripts, although analyses suggest that these reports are no less thorough [26]. However, in the absence of peer review reports (Fig 1B), one method of assessing the reliability of an article is for interested readers or stakeholders to reanalyse the data independently. Unfortunately, we found that many authors offered to provide data only upon request (Fig 1). Moreover, a number of published articles had faulty hyperlinks that did not link to the Supporting information. This supports previous findings of limited data sharing in COVID-19 preprints [27] and faulty web links [28] and enables us to compare trends to the wider literature. It is apparent that the ability to thoroughly and independently review the literature and efforts towards reproducibility are hampered by current data sharing and peer reviewing practices. Both researchers and publishers must do more to increase reporting and data sharing practices within the biomedical literature [15,29]. Therefore, we call on journals to embrace open science practices, particularly with regard to increased transparency of peer review and data availability.

Abstracts represent the first port of call for most readers, usually being freely available, brief, relatively jargon free, and machine readable. Importantly, abstracts contain the key findings and conclusions from an article. At the same time, they are brief enough to facilitate manual analysis of a large number of papers. To analyse differences in abstracts between preprint and paper, we employed multiple approaches. We first objectively compared textual changes between abstract pairs using a computational approach before manually annotating abstracts (Fig 3). Both approaches demonstrated that COVID-19 articles underwent greater textual changes in their abstracts compared to non-COVID-19 articles. However, in determining the type of changes, we discovered that 7.2% of non-COVID-19–related abstracts and 17.2% of COVID-19–related abstracts had discrete, "major" changes in their conclusions. Indeed, 36.1% of non-COVID-19 abstracts underwent no meaningful change between preprint and published versions, although only 24.1% of COVID-19 abstracts were similarly unchanged. The majority of changes were "minor" textual alterations that lead to a minor change or strengthening or softening of conclusions. Of note, 31.9% of changes were additions of new data (Fig 3F) (34.1% COVID-19 and 29.3% non-COVID-19). While previous works have focused their attention on the automatic processing of many other aspects of scientific writing, such as citation analysis [30], topic modelling [31], research relatedness based on content similarity [32], fact checking [33], and argumentative analysis [34], we are not aware of formal systemic comparisons between preprints and published papers that focused on tracking/extracting all changes, with related studies either producing coarse-grained analyses [13] or relying only on derivative resources such as Wikipedia edit history [35] or utilising a small sample size and a single reader [16]. Our dataset is a contribution to the research community that goes beyond the specificities of the topic studied in this work; we hope it will become a useful resource for the broader scientometrics community to assess the performance of natural language processing (NLP) approaches developed for the study of fine-grained differences between preprints and papers. Since our study required the manual collection of abstracts (a process that would be cumbersome for larger sample sizes), this potential would be amplified if increasing calls for abstracts and article metadata to be made fully open access were heeded [29,36] (https://i4oa.org/).

Our findings that abstracts generally underwent few changes was further supported by our analysis of the figures. The total number of panels and tables did not significantly change between preprint and paper, independent of COVID-19-status. However, COVID-19 articles did experience greater variation in the difference in panel and table numbers compared to non-COVID-19 articles. Interestingly, we did not find a strong correlation between how much a preprint changed when published and the number of comments or tweets that the preprint received (Fig 4). This may suggest that preprint comments are mostly not a form of peer review, as supported by a study demonstrating that only a minority of preprint comments are full peer reviews [25]. Additionally, as we have previously shown, most COVID-19 preprints during this early phase of the pandemic were receiving a high amount of attention on Twitter, regardless of whether or not they were published [11].

While our study provides context for readers looking to understand how preprints may change before journal publication, we emphasise several limitations. First, we are working with a small sample of articles that excludes preprints that were unpublished at the time of our analysis. Thus, we have selected a small minority of COVID-19 articles that were rapidly published, which may not be representative of those articles that were published more slowly. Moreover, as we were focusing on the immediate dissemination of scientific findings during a pandemic, our analysis does not encompass a sufficiently long time frame to add a reliable control of unpublished preprints. This too would be an interesting comparison for future study. Indeed, an analysis comparing preprints that are eventually published with those that

never become published would provide stronger and more direct findings of the role of journal peer review and the reliability of preprints.

Furthermore, our study is not a measure of the changes introduced by the peer review process. A caveat associated with any analysis comparing preprints to published papers is that it is difficult to determine when the preprint was posted relative to submission to the journal. In a survey of bioRxiv authors, 86% reported posting before receiving reviews from their first-choice journal, but others report posting after responding to reviewers' comments or after journal rejection [4]. Therefore, the version first posted to the server may already be in response to one or more rounds of peer review (at the journal that ultimately publishes the work or from a previous submission). The changes between the first version of the preprint (which we analysed) and the final journal publication may result from journal peer review, comments on the preprint, feedback from colleagues outside of the context of the preprint, and additional development by the authors independent of these sources. Perhaps as a result of these factors, we found an association between the degree of change and delay between preprint posting and journal publication, although only for non-COVID-19 articles, in agreement with Nicholson and colleagues [14]. COVID-19 articles appear to have consistently been expedited through publication processes, regardless of degree of changes during peer review.

Although we did not try to precisely determine the number of experiments (i.e., by noting how many panels or tables were from a single experimental procedure), this is an interesting area of future work that we aim to pursue.

One of the key limitations of our data is the difficulty in objectively comparing 2 versions of a manuscript. Our approach revealed that computational approaches comparing textual changes at string level do not predict the extent of change interpreted by human readers. For example, we discovered abstracts that contained many textual changes (such as rearrangements) that did not impact on the conclusions and were scored by annotators as having no meaningful changes. By contrast, some abstracts that underwent major changes as scored by annotators were found to have very few textual changes. This demonstrates the necessity that future studies will focus on more semantic NLP approaches when comparing manuscripts that go beyond shallow differences between strings of texts [37]. Recent research has begun to explore the potential of word embeddings for this task (see, for instance, [14]), and Knoth and Herrmannova have even coined the term "semantometrics" [32] to describe the intersection of NLP and scientometrics. Nevertheless, the difficulty when dealing with such complex semantic phenomena is that different assessors may annotate changes differently. We attempted to develop a robust set of annotation guidelines to limit the impact of this. Our strategy was largely successful, but we propose a number of changes for future implementation. We suggest simplifying the categories (which would reduce the number of conflicting annotations) and conducting robust assessments of inter-annotator consistency. To do this, we recommend that a training set of data are utilised before assessors annotate independently. While this strategy is more time consuming (due to the fact that annotator might need several training trials before reaching a satisfying agreement), in the long run, it is a more scalable strategy as there will be no need of a meta-annotator double-checking all annotations against the guidelines, as we had in our work.

Our data analysing abstracts suggest that the main conclusions of 93% of non-COVID-19–related life sciences articles do not change from their preprint to final published versions, with only 1 out of 184 papers in our analysis contradicting a conclusion made by its preprint. These data support the usual caveats that researchers should perform their own peer review any time they read an article, whether it is a preprint or published paper. Moreover, our data provide confidence in the use of preprints for dissemination of research.

## Methods

### Preprint metadata for bioRxiv and medRxiv

Our preprint dataset is derived from the same dataset presented in version 1 of Fraser and colleagues [11]. In brief terms, bioRxiv and medRxiv preprint metadata (DOIs, titles, abstracts, author names, corresponding author name and institution, dates, versions, licenses, categories, and published article links) were obtained via the bioRxiv Application Programming Interface (API; https://api.biorxiv.org). The API accepts a "server" parameter to enable retrieval of records for both bioRxiv and medRxiv. Metadata was collected for preprints posted September 4, 2019 to April 30, 2020 ($n$ = 14,812). All data were collected on May 1, 2020. Note that where multiple preprint versions existed, we included only the earliest version and recorded the total number of following revisions. Preprints were classified as "COVID-19 preprints" or "non-COVID-19 preprints" on the basis of the following terms contained within their titles or abstracts (case insensitive): "coronavirus," "covid-19," "sars-cov," "ncov-2019," "2019-ncov," "hcov-19," and "sars-2."

### Comparisons of figures and tables between preprints and their published articles

We identified COVID-19 bioRxiv and medRxiv preprints that have been subsequently published as peer-reviewed journal articles (based on publication links provided directly by bioRxiv and medRxiv in the preprint metadata derived from the API) resulting in a set of 105 preprint–paper pairs. We generated a control set of 105 non-COVID-19 preprint–paper pairs by drawing a random subset of all bioRxiv and medRxiv preprints published in peer-reviewed journals, extending the sampling period to September 1, 2019 to April 30, 2020 in order to preserve the same ratio of bioRxiv:medRxiv preprints as in the COVID-19 set. Links to published articles are likely an underestimate of the total proportion of articles that have been subsequently published in journals—both as a result of the delay between articles being published in a journal and being detected by preprint servers and preprint servers missing some links to published articles when, e.g., titles change significantly between the preprint and published version [38]. Detailed published article metadata (titles, abstracts, publication dates, journal, and publisher name) were retrieved by querying each DOI against the Crossref API (https://api.crossref.org) using the rcrossref package (version 1.10) for R [38]. From this set of 210 papers, we excluded manuscripts that (1) had been miscategorised by our algorithms as COVID-19 or non-COVID-19; (2) that had been published in F1000Research or a similar open research platform and were therefore awaiting revision after peer review; (3) that were posted as a preprint after publication in a journal; or (4) that did not have abstracts in their published version, e.g., letters in medical journals. This left us with a set of 184 pairs for analysis.

Each preprint–paper pair was then scored independently by 2 referees using a variety of quantitative and qualitative metrics reporting on changes in data presentation and organisation, the quantity of data, and the communication of quantitative and qualitative outcomes between paper and preprint (using the reporting questionnaire; S1 Method). Of particular note, individual figure panels were counted as such when labelled with a letter, and for pooled analyses, a full table was treated as a single panel figure. The number of figures and figure panels was capped at 10 each (any additional figures/panels were pooled), and the number of Supporting information items (files/figures/documents) were capped at 5. In the case of preprints with multiple versions, the comparison was always restricted to version 1, i.e., the earliest version of the preprint. Any conflicting assessments were resolved by a third independent referee.

## Annotating changes in abstracts

In order to prepare our set of 184 pairs for analysis of their abstracts, where abstract text was not available via the Crossref API, we manually copied it into the datasheet. To identify all individual changes between the preprint and published versions of the abstract and derive a quantitative measure of similarity between the 2, we applied a series of well-established string-based similarity scores, already tested for this type of analyses: (1) the python SequenceMatcher (available as a core module in Python 3.8), based on the "Gestalt Pattern Matching" algorithm [24], determines a change ratio by iteratively aiming to find longest contiguous matching subsequence given 2 pieces of text; (2) as a comparison to this open-source implementation, we employed the output of the Microsoft Word version 16.0.13001.20254 track changes algorithm (see details in S3 Method), and used this as a different type of input for determining the change ratio of 2 abstracts. To compute the change ratio of a pair of abstracts, following the Python implementation, the formula is $2^*M/T$, where M is the number of characters in common and T the total number of characters in both sequences. The ratio will span between 1, if the abstracts are identical, and 0 if there is no snippet in common. As Microsoft Word track changes only provides statistics on the characters changed (inserted, removed, etc.) but no information is available on the characters that are in common between 2 abstracts, we derive M by computing the total number of characters in the final abstract minus the characters that have been inserted. Apart from these 2 approaches, there is a large variety of tools and techniques to measure text similarity, especially employing word vector representations (see as a starting point the overview of Task 6 at SemEval 2012 [39], focused on "semantic textual similarity"). However, as these techniques are generally tailored for identifying similarity of "latent" topics more than explicit changes in phrasing, we decided to focus on the 2 approaches introduced above, as we were more familiar with their functionalities and output.

Employing the output of (2), which consisted in a series of highlighted changes for each abstract pair, 4 coauthors independently annotated each abstract, based on a predefined set of labels and guidelines (Table 1, S2 Method). Each annotation contained information about the section of the abstract, the type of change that had occurred, and the degree to which this change impacted the overall message of the abstract. Changes (such as formatting, stylistic edits, or text rearrangements) without meaningful impact on the conclusions were not annotated. For convenience, we used Microsoft Word's merge documents feature to aggregate annotations into a single document. We then manually categorised each abstract based on its highest degree of annotation: "no change" containing no annotations, "strengthening/softening, minor" containing only 1, 1−, or 1+, or "major conclusions change" containing either a 2 or a 3, since only a single abstract contained a 3. See S2 and S3 Tables for a list of representative annotations for each type and all annotations that resulted in major conclusions change. The final set of annotations was produced by one of the authors (MP), who assigned each final label by taking into account the majority position across annotators, their related comments, and consistency with the guidelines.

## Altmetrics, citation, and comment data

Counts of altmetric indicators (mentions in tweets) were retrieved via Altmetric (https://www.altmetric.com), a service that monitors and aggregates mentions to scientific articles on various online platforms. Altmetric provide a free API (https://api.altmetric.com) against which we queried each preprint DOI in our analysis set. Importantly, Altmetric only contains records where an article has been mentioned in at least 1 of the sources tracked; thus, if our query returned an invalid response, we recorded counts for all indicators as 0. Coverage of each

indicator (i.e., the proportion of preprints receiving at least a single mention in a particular source) for preprints were 99.1%, 9.6%, and 3.5% for mentions in tweets, blogs, and news articles, respectively. The high coverage on Twitter is likely driven, at least in part, by automated tweeting of preprints by the official bioRxiv and medRxiv Twitter accounts. For COVID-19 preprints, coverage was found to be 100.0%, 16.6%, and 26.9% for mentions in tweets, blogs, and news articles, respectively.

Citations counts for each preprint were retrieved from the scholarly indexing database Dimensions (https://dimensions.ai). An advantage of using Dimensions in comparison to more traditional citation databases (e.g., Scopus and Web of Science) is that Dimensions also includes preprints from several sources within their database (including from bioRxiv and medRxiv), as well as their respective citation counts. When a preprint was not found, we recorded its citation counts as 0. Of all preprints, 3,707 (14.3%) recorded at least a single citation in Dimensions. For COVID-19 preprints, 774 preprints (30.6%) recorded at least a single citation.

bioRxiv and medRxiv html pages feature a Disqus (https://disqus.com) comment platform to allow readers to post text comments. Comment counts for each bioRxiv and medRxiv preprint were retrieved via the Disqus API service (https://disqus.com/api/docs/). Where multiple preprint versions existed, comments were aggregated over all versions. As with preprint perceptions among public audiences on Twitter, we then examined perceptions among academic audiences by examining comment sentiment. Text content of comments for COVID-19 preprints were provided directly by the bioRxiv development team. Sentiment polarity scores were calculated for each comment on the top 10 most-commented preprints using the lexicon and protocol previously described for the analysis of tweet sentiment.

## Statistical analyses

Categorical traits of preprints or annotations (e.g., COVID-19 or non-COVID-19; type of change) were compared by calculating contingency tables and using chi-squared tests or Fisher exact tests using Monte Carlo simulation in cases where any expected values were <5. Quantitative preprint traits (e.g., change ratios and citation counts) were correlated with other quantitative traits using Spearman rank tests, homogeneity of variance tested for using Fligner–Killeen tests, and differences tested for using Mann–Whitney tests or Kruskal–Wallis for 2-group and more than 2-group comparisons, respectively. All univariate tests were interpreted using a significance level of 0.05, except for pairwise post hoc group comparisons, which were tested using Dunn test adjusting significance levels for multiple testing using Bonferroni correction. Benchmarked statistical power calculations suggested our sample size of $n$ = 184 to detect medium effects with power >0.98 (S1 Appendix).

For multivariate analyses of usage metrics (tweets, citations, and comment counts) and number of authors added, we constructed generalised linear regression models with a log link and negative binomially distributed errors using the function glm.nb() in R package "MASS," v7.3–53 [40]. Negative binomial regressions included automated change ratios of each abstract, manually categorised degree of change to abstracts and figures, COVID-19 status, and delay between preprint posting and publication, adjusting for total time in days each preprint had been online by end of sampling (April 30, 2020). Covariate significance was determined using likelihood ratio tests (LRTs) comparing saturated models with/without covariates. Multicollinearity between covariates was inspected using generalised variance inflation factors (VIFs) calculated using function vif() in R package "car," v3.0–10 [41], ensuring no values were >10. Moreover, 95% confidence intervals (CIs) around resulting rate ratios were calculated using profile likelihoods.

## Parameters and limitations of this study

We acknowledge a number of limitations in our study. First, we analysed only bioRxiv and medRxiv, and many preprints appear on other servers [42]. In addition, to assign a preprint as COVID-19 or not, we used keyword matching to titles/abstracts on the preprint version at the time of our data extraction. This means we may have captured some early preprints, posted before the pandemic, which had been subtly revised to include a keyword relating to COVID-19. Our data collection period was a tightly defined window (January to April 2020 for COVID-19 pairs and September 2019 to April 2020 for non-COVID-19 pairs) meaning that our data suffer from survivorship and selection bias in that we could only examine preprints that have been published and our findings may not be generalisable to all preprints. A larger, more comprehensive sample would be necessary for more conclusive conclusions to be made. Additionally, a study assessing whether all major changes between a preprint and the final version of the article are reflected in changes in the abstract is necessary to further confirm the usefulness of examining variations in the abstracts as a proxy for determining variations in the full text. Furthermore, our automated analysis of abstract changes was affected by formatting-related changes in abstracts, such as the addition or removal of section headers to the abstract. For our manual analysis, each annotator initially worked independently, blinding them to others scoring. However, scores were then discussed to reach a consensus which may have impacted scores for individual pairs. Finally, our non-COVID-19 sample may not be representative of "normal" preprints, as many aspects of the manuscript preparation and publication process were uniquely affected by the pandemic during this time.

## Supporting information

**S1 Fig. Publishing and peer review of preprints during the COVID-19 pandemic broken down by server. (A)** Percentage of COVID-19 and non-COVID-19 preprints published by April 30, 2020. **(B)** Published preprints associated with transparent peer review. **(C)** Data availability for published preprints. **(D)** Change in authorship for published preprints. **(E)** Journals that are publishing bioRxiv preprints. **(F)** Journals that are publishing medRxiv preprints. The data underlying this figure may be found at https://github.com/preprinting-a-pandemic/preprint_changes and https://zenodo.org/record/5594903#.YXUv9_nTUuU. COVID-19, Coronavirus Disease 2019.
(TIFF)

**S2 Fig. Preprint–publication pairs do not significantly differ in the total numbers of panels and tables as broken down by server. (A)** Total numbers of panels and tables in preprints and published articles. Boxplot notches denote approximated 95% CI around medians. **(B)** Difference in the total number of panels and tables between the preprint and published versions of articles. **(C)** Classification of figure changes between preprint and published articles. The data underlying this figure may be found at https://github.com/preprinting-a-pandemic/preprint_changes and https://zenodo.org/record/5594903#.YXUv9_nTUuU. CI, confidence interval.
(TIFF)

**S3 Fig. Granular annotations of changes in abstracts in context of the overall change. (A)** Difflib calculated change ratio for COVID-19 or non-COVID-19 abstracts, based on the overall abstract change. **(B)** Change ratio calculated from Microsoft Word for COVID-19 or non-COVID-19 abstracts, based on the overall abstract change. **(C)** Sum of positive and negative annotations based on the overall abstract change, with colour and label denoting number of abstracts with each particular sum combination. A total of 21 COVID-19 preprints and 35

non-COVID-19 preprints rated "No change" (i.e., sum of positive and negative scores = 0) are not depicted. **(D)** Percentage of annotations in each location within COVID-19 or non-COVID-19 abstracts, based on the overall abstract change. Labels denote absolute number of annotations. **(E)** Percentage of annotations of each type within COVID-19 or non-COVID-19 abstracts, based on the overall abstract change. Labels denote absolute number of annotations. **(F)** Delay (in days) between preprint posting and publication in a journal, based on overall abstract changes. **(G)** Journals publishing COVID-19 preprints, based on overall abstract changes. See S1 Text for key to abbreviated journal labels. The data underlying this figure may be found at https://github.com/preprinting-a-pandemic/preprint_changes and https://zenodo.org/record/5594903#.YXUv9_nTUuU. COVID-19, Coronavirus Disease 2019.
(TIFF)

**S4 Fig. Automated and manually annotated degrees of change to preprints are consistent within infectious disease or epidemiology-related medRxiv preprints (*n* = 57). (A)** Difflib calculated change ratio for COVID-19 or non-COVID-19 abstracts. **(B)** Change ratio calculated from Microsoft Word for COVID-19 or non-COVID-19 abstracts. **(C)** Overall changes in abstracts for COVID-19 or non-COVID-19 abstracts. **(D)** Classification of figure changes between preprint and published articles for COVID-19 or non-COVID-19 abstracts. **(E)** Location of annotations within COVID-19 or non-COVID-19 abstracts. **(F)** Type of annotated change within COVID-19 or non-COVID-19 abstracts. The data underlying this figure may be found at https://github.com/preprinting-a-pandemic/preprint_changes and https://zenodo.org/record/5594903#.YXUv9_nTUuU. COVID-19, Coronavirus Disease 2019.
(TIFF)

**S1 Table. Journals posting preprints from January 1 to April 30, 2020 or September 4, 2019 to April 30, 2020.**
(XLSX)

**S2 Table. Examples of changes in abstracts between the preprint and published version of an article.**
(DOCX)

**S3 Table. All changes in abstracts that resulted in a major conclusion change.**
(DOCX)

**S1 Material. Abstract annotations utilised for the analysis in this study.**
(DOCX)

**S2 Material. Nonresolved abstract annotations provided for NLP researchers.** NLP, natural language processing.
(DOCX)

**S1 Method. Questionnaire used for assessing manuscript metadata, panels, and tables.**
(PDF)

**S2 Method. Rubric for annotating abstracts.**
(DOCX)

**S3 Method. Protocol for comparing and extracting annotations from Word files.**
(DOCX)

**S1 Text. Key for journal abbreviations from Fig 2D and 2E and S3G Fig.**
(DOCX)

**S1 Appendix. Illustrative power calculations for chi-squared and correlation tests.**
(DOCX)

## Author Contributions

**Conceptualization:** Liam Brierley, Federico Nanni, Jessica K. Polka, Gautam Dey, Máté Pálfy, Nicholas Fraser, Jonathon Alexis Coates.

**Data curation:** Liam Brierley, Jonathon Alexis Coates.

**Formal analysis:** Liam Brierley, Federico Nanni, Jessica K. Polka, Gautam Dey, Máté Pálfy, Nicholas Fraser, Jonathon Alexis Coates.

**Investigation:** Liam Brierley, Federico Nanni, Jessica K. Polka, Gautam Dey, Máté Pálfy, Nicholas Fraser, Jonathon Alexis Coates.

**Methodology:** Liam Brierley, Federico Nanni, Jessica K. Polka, Gautam Dey, Máté Pálfy, Nicholas Fraser, Jonathon Alexis Coates.

**Project administration:** Jonathon Alexis Coates.

**Resources:** Jessica K. Polka, Jonathon Alexis Coates.

**Software:** Liam Brierley, Nicholas Fraser.

**Supervision:** Jonathon Alexis Coates.

**Validation:** Jessica K. Polka, Jonathon Alexis Coates.

**Visualization:** Liam Brierley, Federico Nanni, Jessica K. Polka, Nicholas Fraser, Jonathon Alexis Coates.

**Writing – original draft:** Liam Brierley, Federico Nanni, Jessica K. Polka, Gautam Dey, Máté Pálfy, Nicholas Fraser, Jonathon Alexis Coates.

**Writing – review & editing:** Liam Brierley, Federico Nanni, Jessica K. Polka, Gautam Dey, Máté Pálfy, Nicholas Fraser, Jonathon Alexis Coates.

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
