## [Editor Report · Decision Letter 0]

21 May 2021

Dear Dr Coates, 

Thank you for submitting your manuscript entitled "Preprints in motion: tracking changes between posting and journal publication" for consideration as a Meta-Research Article by PLOS Biology.

Your manuscript has now been evaluated by the PLOS Biology editorial staff, as well as by an academic editor with relevant expertise, and I'm writing to let you know that we would like to send your submission out for external peer review.

Please re-submit your manuscript within two working days, i.e. by May 25 2021 11:59PM.

Kind regards,

Roli Roberts

Roland Roberts

Senior Editor

PLOS Biology

rroberts@plos.org

---

## [Decision Letter · Decision Letter 1]

16 Jul 2021

Dear Dr Coates,

Thank you very much for submitting your manuscript "Preprints in motion: tracking changes between posting and journal publication" for consideration as a Meta-Research Article at PLOS Biology. Your manuscript has been evaluated by the PLOS Biology editors, an Academic Editor with relevant expertise, and by three independent reviewers.

You'll see that two of the reviewers are positive about your study, and although reviewer #2 questions the magnitude of the advance, they do so constructively, and we think that there is enough enthusiasm elsewhere to invite a revision. Between them, reviewers #2 and #3 request a significant number of additional analyses. Reviewer #1 and the others also have substantial textual and/or presentational requests.

In light of the reviews (below), we are pleased to offer you the opportunity to address the comments from the reviewers in a revised version that we anticipate should not take you very long. We will then assess your revised manuscript and your response to the reviewers' comments and we may consult the reviewers again.

We expect to receive your revised manuscript within 2 months.

**IMPORTANT - SUBMITTING YOUR REVISION**

*Resubmission Checklist*

*Published Peer Review*

*PLOS Data Policy*

*Blot and Gel Data Policy*

Sincerely,

Roli Roberts

Roland Roberts

Senior Editor

PLOS Biology

rroberts@plos.org

REVIEWERS' COMMENTS:

Reviewer #1:

[identifies himself as Ross Mounce]

This manuscript 'Preprints in motion: tracking changes between posting and journal publication' presents an analysis of (only bioRxiv and medRxiv) 14,812 preprints that had been posted between 1st January and 30th April 2020. They then identify which of these subsequently can be identified as published in a journal and classify them as either COVID-19 or not COVID-19 related. They then attempt to compare n=105 COVID-19 related preprint-paper pairs with n=105 not COVID-19 related preprint-paper pairs to discern changes between preprint and journal-published versions, and to see if there are any differences between the COVID-19 related and not COVID-19 related sets. They focus on textual changes in abstracts, article level metrics, and on the number of figures and tables between preprint and journal-published versions of a work. The results are newsworthy e.g. "the main conclusions [in abstracts] of 94% of non-COVID-related life sciences articles do not change from their preprint to final published versions".

I think it's an interesting manuscript worthy of publication in PLOS Biology after a few relatively minor revisions.

I thank for the authors for putting the underlying code and data on github (https://github.com/preprinting-a-pandemic/preprint_changes) AND for preserving a proper archive snapshot of that material on Zenodo. Great practice.

**Comments on the abstract of this manuscript**

In my opinion the abstract of this manuscript contains a debatable assertion which I strongly disagree with: the last line states "…but the majority of these changes do not reverse the main message of the [each?] paper."

I simply don't perceive that research necessarily has or should have a singular 'main message'. Research articles are complex things. Some can be hundreds of pages long. Some can be entirely descriptive e.g. taxonomic papers. The idea that research papers should be massaged to have one clear takeaway 'main message' is harmful to robust reporting of reproducible research, and it encourages questionable research practices i.e. to 'sharpen the narrative'. At the very least, I think the authors of this manuscript should just talk more about "conclusions" (plural!) and very specifically in a pluralised way. Put another way, can you recall reading a paper which performs and reports only one test? I cannot, if they exist they are super-rare. Most papers test a multiplicity of things and it is up to the reader to decide which of the tests reported are the 'main' for them and their purposes. 

Likewise I also wonder about the word 'reverse' in abstract. What does 'reversal' mean in this context? It appears to be assumed that all readers will have a shared understanding of exactly what this means. It seems to imply binary states to me. But I'm not sure that all conclusions from all possible research tests are necessarily so binary.

For the sake of precision I must insist that the abstract should also make it clear that only bioRxiv and medRxiv preprints were sampled in this analysis. These preprint servers are not the only recognised preprint servers with COVID-19 related content out there. Moreover there are plenty of researchers e.g. in the phylodynamics community who aren't even using formal preprint servers and instead are just putting up manuscript drafts (?are these preprints?) on https://virological.org/

I also note that no mention of the choice/restriction to just bioRxiv and medRxiv prerprint servers is given in the "Parameters and limitations of this study" section. This is very much an arbitrary parameter choice and limitation to the study and so should also be discussed in that section too, even if just a few words given to acknowledge it.

**Comments on methods**

Line 418-419 In the context for recent initiatives such as I4OA (initiative for open abstracts), it might be interesting to report the number of abstracts out of 200 which were not available via the Crossref API.

Statistical reporting: for tests I see that degrees of freedom and p-values are reported, but there are a few things that could have been made more explicit

e.g. for the following line:

"…a statistically significant increase compared to the 3.0% of non-COVID-19 preprints that were published (Chi-square test; χ2 = 6.77, df = 1, p=0.009) (Fig. 1A)."

I guess it is implicit that your chosen significance level (α) is 0.05? Perhaps explicitly state it?

What are the sample sizes for this test (and other tests reported in the manuscript)? I don't doubt it is reported somewhere in the manuscript but since the sample sizes vary depending on which exact test performed, the specific inclusions and exclusions, I do think it is good practice to report the sample sizes directly with the reported results so there is no chance of confusion / maximum clarity. Figure 1A doesn't help much because it also only has percentages - I presume the Chi-square test here was performed on the actual frequencies and not percentages? Perhaps put the absolute numbers on the figure, not just percentages alone?

Could you elaborate on the experimental design a little more from a statistical power perspective? i.e. does the seemingly arbitrary choice of sample sizes of 99 non-Covid and 101 Covid matched preprint-paper pairs, give enough statistical power to draw conclusions from? Put your calculations in the manuscript to better justify the sample sizes used. Post hoc power analysis is better than no power analysis whatsoever, although of course in an ideal world it might have been nice to preregister this research. Consider preregistering your experimental design and hypotheses to test perhaps for next time?

Firstly I assumed the sample sizes were 105 (covid) + 105 (non-covid), but then I saw it was 101 (covid) + 99 (non-covid) =200

But then I see this written at line ~370 "one out of 185 papers in our analysis reversing the conclusion made by its preprint", 185 is not 200. Please explain/clarify that denominator as it is a super interesting statistic but not one I trust much until I better understand its provenance.

To improve reproducibility, please state the exact versions of the software you used e.g. for rcrossref . This information was not found in either the manuscript main text, nor in the reference list where it is merely given as: 

35. Chamberlain S, Zhu H, Jahn N, Boettiger C, Ram K. rcrossref: Client for Various "CrossRef" "APIs." 2020. Available: https://CRAN.R-project.org/package=rcrossref

the python SequenceMatcher is contained within the python difflib module, so you should probably cite and give the version information for difflib

likewise more precise information on the software used to implement the Microsoft Word track changes algorithm should be given (Microsoft Word? if so which version!)

Why were these two specific algorithms used (sequencematcher and MS-Word track changes), how do they compare to the wide panoply of other algorithms used for text similarity comparison out there? This is rather undiscussed... Please cite some more papers to acknowledge the other algorithms and research on text similarity out there, and perhaps feel free to admit the choice was just convenience and familiarity (if it was. Alternatively convince us _why_ these two are the best to use for comparing abstract texts).

**Comments on "Parameters and limitations of this study" **

Great that you have such a section in your manuscript. Not everyone does.

I am surprised that there are no remarks given in this section as to the correspondence between mere abstracts, and the entire full text of a published research article. Much of the language and methodology in this research revolves around abstracts being representative of "main conclusions". As I have previously said research articles are long and complex things and by doing analyses of abstracts and the number of figures/tables you're really only skimming the surface of them. The analysis of abstract text-only was a design choice, and it should be discussed that this manuscript _could_ have used the full text of the research paper like the cited Nicholson et al "Linguistic Analysis of the bioRxiv Preprint Landscape" preprint. There are factors that make full-text analysis difficult e.g. that not all research papers are openly available, but interestingly that is also the case with abstracts (not all are openly available at CrossRef). I would like to see a little bit of acknowledgement of uncertainty around exactly whether abstracts always contain all the "main conclusions", are abstracts always well-written? Are all hypotheses tested in a research paper ALL reported in an abstract (no, they are not). There is certainly information loss occurring by only using abstracts is what I'm saying, and the "main" -ness of conclusions represented within the tight word-limit confines of an abstract is subjective.

Consider a discussion of the text similarity algorithms used, and those not used here too.

 All in all this is great work and most of my comments are mostly just trying to encourage better precision and clarity.

Reviewer #2:

Learning more about preprints during the COVID-19 pandemic seems to be a worthwhile endeavor. Whether the advancements of this manuscript over prior or related research are sufficiently noteworthy for publication might lead to subjective answers - and either view would seem possible to me. In my opinion, improved statistical rigor (improved reporting on uncertainty and considering non-COVID-19 literature from same time window as COVID-19 literature) and a more conservative interpretation of their findings would seem essential to prevent misreading several of their findings and to ensure that claims are valid. 

Personally, I was most intrigued by the absence of large effects, which could hint at preprints or pre-publication review doing little to better the quality of research articles. Along this line, I was left wondering whether the study suggests that also the peer review through journals has little effect on the published work aside from possibly acting as a filter for publication. Limiting the appeal of such an interpretation, the authors however discuss caveats that caution against considering their study as a measure of changes introduced by the peer review process.

In my view the readability and usefulness of the manuscript could be extended greatly by including confidence intervals to the figures (e.g.: error bars to barplots, and notches to boxplots). The manuscript of Polka et al. inherently inspects a finite site of preprints, and - for understandable reasons - sometimes investigates only a subset of these. While I do not see this as an inherent limitation of the study given its targeted scope, small numbers will reduce statistical power. Currently, it is not clear from figures (or text) to which extent reports on presence and absence of a statistically significant different are due to effect sizes or other factors that will result in uncertainty in the analysis. Similarly, when statistical differences are claimed, it is not clear if the effect sizes could possibly be quite small. Confidence intervals for count-based data could possibly be obtained by bootstrapping, and some visualization software packages such as Python's Seaborn would already provide that option. 

I believe that the text would need to be more careful when presenting their study as evidence against the association of preprints with "low quality", although I believe that it is fair to conclude from their data that most preprints have a similar quality to most published articles. First, the rejection of manuscripts for publication could serve as a filter against "low quality", and the present study does not investigate unpublished preprints. Second, journal reputation might signal the extent of quality - whereas this indicator is missing for preprints. Third, media coverage is not equally concerned with all preprints - thus leaving the possibility that a small number of "low quality" preprints could disproportionally affect public perception about scientific topics, including COVID-19.

The text should be more explicit on the time windows considered for preprint publication dates and the last date considered for journal publication. Specifically, Supplemental Figure 3F suggests that the time window could be larger than the 4 months stated in the text. 

Further, it is my understanding of the methods section that the non-COVID-19 literature included the time frame of the COVID-19 literature, but also an equally long duration that preceded the onset of the time frame of the COVID-19 literature. Given their own data of Fraser et al., one would anticipate that the preprints in those two different observational periods behaved differently. Thus, the manuscript plausibly does not allow to conclude whether claimed differences between "COVID-19" and "non-COVID-19" are due to COVID-19 or due to broader observational window of "non-COVID-19" that additionally includes times prior COVID-19. If different time windows were used for "COVID-19" and "non-COVID-19" I would suggest redoing the analyses with a consistent time window.

Despite related work from Carneiro et al. 2021, it is unclear to which extent "data availability" is a more general surrogate for "quality". Plausibly, many publications could be seen as being high quality, even if data is not available (e.g.: see Stodden et al. 2018 on study of Science, a journal enjoying a high reputation). I would advise for a more specific or weaker language.

When referring to comparisons, adding significance values to text would be necessary to see if claims are supported. One instance where it seems critical for the understanding is line 124f where the statement of specific type of change (14% vs 5%) does - in contrast to the opening statement of any change - not provide statistical support. 

Throughout the manuscript, it is unclear whether differences between COVID-19 and non-COVID-19 could be partially explained by research fields - or bioRxiv categories as a surrogate for the latter. Maybe it is possible to address this briefly for 1-2 claims, or by doing a supplemental analysis based on a single bioRxiv category, or by discussing it more explicitly when interpreting the data.

The section starting on line 130 does not seem necessary and lacks statistical support and does not allow readers to estimate, whether effects partially stem from different volumes of COVID-19 and non-COVID-19 research. Possibly the number of articles per journal is too low beyond an anecdotal conclusion that a wide variety of journals has published ~2 manuscripts that were also preprints. In contrast, the statement on clinical or multidisciplinary journals seems intriguing, but would need to be supported by a dedicate analysis (e.g.: classifying and pooling journals, and adding statistical analysis).

The statement "Moreover, for individual preprint-published pairs, we found that there was greater variation …. (Figure 2B)" does not seem evident from the referenced panel - especially as the span is greater for non-COVID-19 in bioRxiv and the shown distribution seems to have gone some smoothing. In addition to providing statistical support, I would suggest visualizing the absolute delta of panels and tables as a boxplot that also shows the interquartile range (or even better: a letter-style plot that shows several quantiles, or as a cumulative / survival plot).

Panels Figure 2D, E do not seem to be referenced or discussed in the text.

It is unclear whether the computational approaches to measure textual differences would be affected by text length or the adoption of "Background/Results/Discussion" subheadings that exist within the abstracts of a minority of journals. 

The authors do not provide a quantification into whether the results of Figure 3A would be statistically significant.

The caption of figure 3 reads "preprint-publication abstract pairs are not significantly different", but panel 3A and the accompanying text "we found that COVID-19 abstracts had more changes than non-COVID-19 abstracts" seem to suggest the presence of significant differences.

The data shown in Supplemental Figure 3F toward delays between article publication days and preprints sometimes shows a delay in the proximity of ~10 days, which is a fast turn-around time for peer review and editorial work. This invites the question whether - as the authors briefly discuss in the discussion sections - the findings of Polka et al. truly reflect on the course between preprints and journal publication, or whether there are many manuscripts that only get posted as preprints once the journal signaled acceptance. While the authors might not be able to resolve this, I would advise to present this alternative reading more prominently and/or to try to include an analysis to address above.

Figure 4 A-D are very exciting as they seem to defy the expectation that twitter and comments on preprint servers would provide feedback used by the authors. In my view this is the most novel and intriguing finding of the manuscript - even though it is discomforting as it implies that pre-publication discussions do little to improve the quality of science. 

As Figure 4 A-D run against plausible anticipations or hopes toward preprints, a slightly more extended analysis would seem useful to contextualize their findings. First, what is the difference between COVID-19 and non-COVID-19 manuscripts. Second, how would preprints that are not accepted for publication compare.

The findings of Figure 4E on the relation between citations and extent of changes make me wonder, whether they also hold when controlling for the month of the posting and/or publication.

Extending beyond Figure 4E and returning to earlier panels on main findings (Figure 1D, 2C, 3C), it would be interesting to see data broken down by months (e.g.: was initial month of pandemic very different from later ones …).

The discussion section mentions the "beneficial effect upon preprints". While one might spot these by comparing effect- and effect+ in Figure 3F, effect sizes are small and no statistics are provided. I believe it would be safer, and more justified to the manuscript, to refer to other publications in case that the authors want to discuss positive aspects of preprints.

Along this line of reasoning, I was most impressed by the apparent absence of major effects of preprints and peer-review toward the final accepted manuscript and the absence of detectable associations between comment/tweet numbers and the extent of changes. Although this might be seen as a negative finding, and may not be the findings that many scientists (and peer reviewers) would personally like to have seen, I would enjoy a further discussion of this point and would be curious about the authors' thoughts.

According to methods section, the maximal number of figures and panels was capped at 10 and the number of supplemental materials capped at 5. While understandable, this could lead to the share of preprints without change having been overestimated (e.g.: if lengthy preprints with would undergo changes).

Other: 

Line 95: The sentence on "At the time of our data collection in May 2020, …" is a bit unclear regarding which months were considered for COVID-19 and non-COVID-19 literature, and might even be misleading if my reading of the methods section dedicated to non-COVID-19 literature was correct.

Panel 1B could go further. The main impression from the panel is that few journals do open peer review. This might not be surprising. However, the figure (and text) does not proceed to investigate two points that would seem to add to the main narrative. First, are there statistical differences between COVID-19 and non-COVID-19 (e.g.: the results could be alternatively visualized to show that in non-COVID-19 transparent peer review is ~5 times higher than in COVID-19, which might have interesting/discomforting implications on openness or rigor of COVID-19 literature). Second, although only a subset of articles has transparent peer review, the authors do not report on whether that reduced number would still have sufficed to address their initial goal of estimating quality. Additionally, several publishers offer peer reviewing data to scientists performing meta-research - which might have enabled the authors to proceed toward their investigation of "quality" even if the public data was scarce. 

The statement on line 160 might not be "surprising" given Adams et al. 2021, Nature.

Neither text nor supplemental method 3 seem to indicate what "Microsoft word track changes algorithm" measures. Especially, it is unclear to me how to interpret scores above 1.0 (e.g.: would 1.1 and 0.9 be equally close to large rewrites as equidistant from a value of 1.0, as suggested on line 188).

Line 177: "drastically re-written" could possibly be toned down, since the reference frame for what constituted drastic re-writes might be subjective and difficult to define.

Panel 3D is very difficult to read as most information is close to 0 while the text intends to convey that there are some manuscripts that mainly undergo positive or negative changes. I am unsure whether the latter warrants a main figure. Would suggest testing (and maybe include if clearer) an alternate visualization, where x-axis is given by total number of positive and negative scores, and y-axis shows in a boxplot for each preprint-publication pair, which share of the score is falls onto positive changes. Thus, one could see if there is generally more variability, or if there also is a systematic trend.

While the paragraph dedicated to Supplemental Fig 3G is somewhat interesting, the absolute number of articles is small (generally ranging between 1 and 5). Consequently, it is difficult to conclude whether the statements from the text refer to anything that is specific to preprints, or if they merely reflect on different journals publishing a different number of articles (e.g.: finding multiple articles from PLOS ONE might indicate that PLOS ONE publishes more articles than other journals).

While the authors' attempts to use controlled criteria for the manual scoring of changes are laudable, it is unclear, whether additional measures had been taken to avoid human biases (e.g.: blinding annotators to prior results, or possible scope of publication) - especially as the discussion section notes that "different assessors may annotate changes differently". While I do not intend to suggest new analysis such a limitation could be noted in the main text. 

To better connect Figures 3 and 4, it would be nice if an analysis along Figure 4 would also be repeated in supplemental figures for the metric of change presented in Figure 3A. Given the ordinal nature of the metric of Figure 3A, the authors could also do a scatter plot or provide a Spearman correlation (e.g.: do articles with larger textual change also get a higher number of tweets….). Adding such an analysis might also allow for a stronger discussion of differences between computational and expert-supervised approaches to identify changes.

The lengthy discussion of data availability (and using it as a readout of "quality") seems peripheral to the manuscript and its findings.

The statement on "this potential would be amplified if increasing calls for abstracts and article metadata to be made fully open access" does not flow from the manuscript and falls short to recognize that several publishers will provide researchers of science with additional data around their publications and peer review process. Hence open access is not strictly needed (although it would be convenient). 

The discussion refers to "our approach revealed that computational approaches comparing textual changes at string-level are insufficient", whereas the argument to be drawn from the discussion is that computational and non-computational approaches can differ (hence either could be insufficient or complementary). Moreover, Figure 3 A-C show that overall, they don't differ in the resulting main conclusions. 

The authors' scoring criteria for change considers research claims, but - in possible opposition to the computational approaches - not the clarity of the text or its adjustment to a specialized group of readers. Improved communication could be seen as one of several qualities of manuscripts. 

Computational approaches to identify similarity between abstracts could be extended toward the recognition and differences in key concepts. While I'm not suggesting new work, there might be a way to add a half-sentence to the discussion - possibly referring to Nicholson et al. 2021 .

The methods section lists the transformation of "fever" to "high fever" as as "substitution of a noun". While I see the underlying rationale for distinguishing "fever" and "high fever", I am unsure whether - grammatically - above is an example of the substitution of a noun. Instead, it appears to me that above is the addition of an adjective as an attribute to a noun, and might even fall into the effect+ category as "high fever" (which would still be a fever) would remove fever with lower temperatures.

Quantifying whether titles changed could allow for additional comparisons to existing research into changes of preprints (Lin et al., 2020, Scientometrics).

While the authors follow the reasonable reductionistic assumption that peer-review and preprints have the potential to improve manuscripts, Crane, 1967, The American Sociologist, discusses the more nuanced possibility that peer review serves to make research more conservative. Maybe the discussion section could propose ways how their strategy could be applied to investigate above.

Reviewer #3:

This study asks an important and salient question in scientific publishing: (how) do preprints change between their initial release on a preprint server and their eventual publication in a peer-reviewed journal? This has major implications for how we assess the quality of preprinted research, the degree to which we should depend on pre-publication peer review, and the value we assign to peer review as an academic practice, along with peer-reviewed journals as a whole. The paper is well-written and the authors have clearly gone to great lengths to curate and analyze this dataset in a way that enables an extremely thorough assessment of the changes preprints undergo on their journey to peer-reviewed publications.

My only major criticism of this study is that the non-COVID-19 articles included in the analyses are not necessarily representative of the "baseline" for how articles change between preprint and peer-reviewed publication, despite the authors seeming to make this generalization throughout the paper. Many of the non-COVID-19 articles posted during this time were likely impacted by the pandemic through lockdowns resulting in lost time in the lab, shortages and diversions of resources (including human, monetary, and lab supplies/equipment), and the broad psychological effects of the pandemic. In addition to many journals expediting peer review of COVID-19 articles, they also often generous extensions to reviewers and authors of non-COVID-19 submissions and reviewers were encouraged to be more lenient in requesting new experiments, etc. that could not feasibly be completed due to lockdowns. These factors undoubtedly could have an impact on many of the metrics evaluated here (e.g., time to publication, evidence of additional experiments, etc.).

Ideally, this study would be revised throughout to consider a third group of preprints posted before the pandemic for comparison, e.g., those posted between Jan-Apr 2019, but I recognize that this is likely a massive amount of work and would only consider it worthwhile if the authors feel confident it could be completed within a typical response window. Instead, I would suggest the authors carefully go through the manuscript and ensure that they are not overstating the scope of their study and conclusions (the title may also need to be revised to reflect this limited scope). Alternatively, I wonder if the non-COVID-19 preprints could be analyzed in separate groups, one from 1 Jan - 11 Mar, 2020 (before the WHO announced the pandemic) and the other from 11 Mar - 30 Apr 2020.

Minor comments:

Line 102-103: I wouldn't necessarily claim that medical/epidemiological medRxiv preprints are "more relevant to the pandemic" than, say, bioRxiv preprints in evolutionary biology attempting to determine the origin of SARS-COV-2.

Line 112-114: While I fully agree that requiring underlying data to be accessible is an important mechanism to promote transparency and establish confidence in published research, this section seems to conflate the actual quality of the research with the ability to assess that quality because data are available. A reduction in data availability between preprint and peer-reviewed forms of a paper is not evidence of a reduction in quality (and vice versa). Further, given that data availability (or lack thereof) seems to be mostly a matter of different standards/expectations between fields/journals, I'm not convinced changes to data availability are a meaningful measure of differences between COVID-19 and non-COVID-19 preprint-publication pairs.

Line 122: This could use a citation.

Line 127: I'd also be interested to see, among the ~25% of papers with authors added, how many authors were added, and if this also correlates with time to publication and/or significant changes to the text of the paper.

Line 177: Can you include the exact value of "sizable number" in the text?

Line 220-222: Based on conversations with colleagues involved in COVID-19 research, my impression is that the tendency of these papers to update with larger sample sizes ultimately boils down to reviewers/editors requesting the analyses be repeated with new additional data (especially among epidemiological studies), due to the pandemic's rapid progression. Could this be broken down by subject area? If there are differences, this could be useful to COVID-19 researchers in isolating which subject areas are likely to request additional data in peer review and preemptively preparing for such updates.

Line 237-239: it seems highly counterintuitive that non-COVID-19 articles with major changes did not experience significantly longer delays than those with no change. Is this simply due to lack of power because there were so few articles with major changes?

Line 274-275: it should be trivially easy to test with regression or ANOVA whether the increased citations of preprints undergoing discrete changes is in fact associated with a lengthier peer review time (or, more precisely, time between preprint and peer-reviewed publication).

---

## [Editor Report · Decision Letter 2]

22 Oct 2021

Dear Dr Coates,

Thank you for submitting your revised Meta-Research Article entitled "Preprints in motion: tracking changes between preprint posting and journal publication during a pandemic" for publication in PLOS Biology. The Academic Editor and I have now assessed your revisions and responses to the reviewers.

Based on this assessment, we will probably accept this manuscript for publication, provided you satisfactorily address the following data and other policy-related requests.

IMPORTANT: Please address the following:

a) Please could you simplify the title of your article? We suggest either "Tracking changes between preprint posting and journal publication during a pandemic" or the more general "Tracking changes between preprint posting and journal publication."

b) Attend to my Data Policy requests below; specifically, we will need the numerical values underlying Figs 1ABCDE, 2ABCDE, 3ABCDEF, 4ABCDEF, S1ABCDEF, S2ABC, S3ABCDEFG, S4ABCDEF. My understanding is that these can be derived from your Zenodo/Github deposition. Please can you clarify that this is the case; if so, cite the URLs clearly in each main and supplementary Figure legend (e.g. "The data underlying this Figure may be found at https://..." or "The data and code needed to generate this Figure may be found at https://...".

We expect to receive your revised manuscript within two weeks. 

*Published Peer Review History*

*Early Version*

Sincerely,

Roli Roberts

Senior Editor,

rroberts@plos.org,

PLOS Biology

DATA POLICY:

Regardless of the method selected, please ensure that you provide the individual numerical values that underlie the summary data displayed in the following figure panels as they are essential for readers to assess your analysis and to reproduce it: Figs 1ABCDE, 2ABCDE, 3ABCDEF, 4ABCDEF, S1ABCDEF, S2ABC, S3ABCDEFG, S4ABCDEF. NOTE: the numerical data provided should include all replicates AND the way in which the plotted mean and errors were derived (it should not present only the mean/average values).

DATA NOT SHOWN?

---

## [Editor Report · Decision Letter 3]

28 Oct 2021

Dear Dr Coates,

On behalf of my colleagues and the Academic Editor, Ulrich Dirnagl, I'm pleased to say that we can in principle accept your Meta-Research Article "Tracking changes between preprint posting and journal publication during a pandemic" for publication in PLOS Biology, provided you address any remaining formatting and reporting issues. These will be detailed in an email that will follow this letter and that you will usually receive within 2-3 business days, during which time no action is required from you. Please note that we will not be able to formally accept your manuscript and schedule it for publication until you have any requested changes.

PRESS: We frequently collaborate with press offices. If your institution or institutions have a press office, please notify them about your upcoming paper at this point, to enable them to help maximise its impact. If the press office is planning to promote your findings, we would be grateful if they could coordinate with biologypress@plos.org. If you have not yet opted out of the early version process, we ask that you notify us immediately of any press plans so that we may do so on your behalf.

Sincerely,

Roli 

Roland G Roberts, PhD 

Senior Editor 

PLOS Biology

rroberts@plos.org